# Treatment of Blepharospasm and Oromandibular Dystonia with Botulinum Toxins

**DOI:** 10.3390/toxins12040269

**Published:** 2020-04-22

**Authors:** Travis J.W. Hassell, David Charles

**Affiliations:** Department of Neurology, Vanderbilt University Medical Center, 1161 21st Avenue South, Suite A-1106 MCN, Nashville, TN 37232, USA; david.charles@vumc.org

**Keywords:** blepharospasm, oromandibular dystonia, Meige syndrome, botulinum toxin, OnabotulinumtoxinA, AbobotulinumtoxinA, IncobotulinumtoxinA, RimabotulinumtoxinB, DaxibotulinumtoxinA

## Abstract

Blepharospasm and oromandibular dystonia are focal dystonias characterized by involuntary and often patterned, repetitive muscle contractions. There is a long history of medical and surgical therapies, with the current first-line therapy, botulinum neurotoxin (BoNT), becoming standard of care in 1989. This comprehensive review utilized MEDLINE and PubMed and provides an overview of the history of these focal dystonias, BoNT, and the use of toxin to treat them. We present the levels of clinical evidence for each toxin for both, focal dystonias and offer guidance for muscle and site selection as well as dosing.

## 1. Introduction

Cranial dystonias are hyperkinetic movement disorders manifested by abnormal, involuntary movements of cranial muscles, characterized by intermittent or sustained muscle contractions. Two of the most common focal cranial dystonias are blepharospasm and oromandibular dystonia (OMD). Blepharospasm most commonly includes the bilateral orbicularis oculi muscles but may rarely begin unilaterally [1]. OMD involves oropharyngeal and jaw muscles with variable presentations of jaw, mouth, tongue, and lower facial movements. Meige syndrome is a variant that describes the co-existence of blepharospasm and oromandibular dystonia [2]. 

This work presents an important background and epidemiology of cranial dystonias and summarizes the level of evidence for each botulinum neurotoxin (BoNT) for blepharospasm, OMD, and Meige syndrome. We discuss technical aspects of BoNT use and provide guidance on muscle selection, optimal injection sites, and dosing. 

## 2. Literature Review Methods

An electronic literature search of both MEDLINE and PubMed databases was conducted in January 2020 to identify relevant literature regarding the use of botulinum toxin type A (BoNT/A) and type B (BoNT/B) for blepharospasm, oromandibular dystonia, and Meige syndrome. All botulinum toxin strains were considered. Special attention was placed on the meta-analyses and randomized controlled trials, in order to gather more comprehensive data that highlight the similarities and differences between toxins. 

The following search strategy combining Medical Subject Headings (MeSH) as follows: (Blepharospasm (MeSH) OR Oromandibular dystonia (MeSH) OR Meige syndrome OR Meige OR Dystonia OR Cranial dystonia OR Hemi-facial spasm) AND (Botulinum toxin OR Botulinum toxin Type A OR Botulinum toxin Type B)

This search returned 2200 results. The search was then narrowed to 1629 results by restricting to human data in the English language, with no time range. By giving preference to original or seminal articles, meta-analyses, randomized controlled trials, multi-center studies, and systematic reviews the results included 1525 articles. From this pool, only journal articles were considered, which reduced the results to 717. These were reviewed for relevance to the core topic of BoNT therapy for blepharospasm, oromandibular dystonia, and Meige syndrome. When appropriate, seminal journal articles that were not in the results listed above were included to provide a more comprehensive background. Figure 1 depicts this search process.

## 3. Blepharospasm and Oromandibular Dystonia: Classification, Epidemiology, and Clinical Description

### 3.1. Classification

Dystonia is defined as abnormal involuntary muscle contractions that can be intermittent or sustained and cause an abnormal posture that is often characterized by a twisting or turning nature. These movements are often patterned or repetitive when dynamic. Muscle contractions can be spasmodic or tonic, dynamic or fixed, or any combination that causes an abnormal posture of the affected body region. Dystonia is further categorized by two main criteria: (1) Clinical features that include age at onset, body distribution (focal, segmental, and generalized), temporal pattern, and associated features or neurologic disorders such as parkinsonism; (2) etiology: Inherited, acquired, or secondary. Idiopathic blepharospasm and oromandibular dystonia are examples of focal dystonia as they involved only one body region. The combination of the two is called Meige syndrome which is a segmental cranial dystonia because it involves two contiguous cranial-nerve innervated body regions [3]. Secondary causes of dystonia include medications, trauma, stroke, demyelination, hypoxia, kernicterus, normal pressure hydrocephalus, or tumor in the basal ganglia or brainstem. [4,5,6]. Medication-induced secondary dystonia is most often due to dopamine receptor modulation and is termed tardive dystonia. Some examples include dopamine blocking anti-emetics, neuroleptics for psychiatric disorders, and dopamine receptor agonists such as levodopa.

### 3.2. Epidemiology

The phenotypic presentation of each focal dystonia correlates to the age of onset [7]. Blepharospasm and oromandibular dystonia (55.7 years) are more likely to be present 15 years later than other focal dystonias such as writer’s cramp (38.4 years) or cervical dystonia (40.8 years) [7]. There are also differences between sexes with a trend toward female predominance. The female-to-male ratio ranges from 1.6:1 to 3.3:1 [8]. 

The overall prevalence of cranial and focal dystonias is not clearly defined. Reported prevalence rates vary widely from 50 cases per million for early-onset primary dystonia and 30 to 7320 cases per million for late-onset primary dystonia [9]. A regional U.S. prevalence study estimated a range of 13 to 130 cases per million for blepharospasm and 69 per million for OMD [10]. A multi-center European prevalence study across eight countries showed the prevalence of primary focal dystonia to be 117 per million with blepharospasm accounting for 36 per million (95% CI 31–41) in Europe in 2000 [11].

### 3.3. Clinical Descriptions

#### 3.3.1. Blepharospasm

Blepharospasm is the most common focal cranial dystonia and classically involves bilateral orbicularis oculi muscles. When blepharospasm occurs in isolation it is termed essential blepharospasm. However, blepharospasm is often associated with other focal dystonias, such as facial, oral, lingual, or oromandibular dystonias. Oromandibular dystonia is the most commonly associated dystonia with blepharospasm. 

Blepharospasm most commonly manifests as bilateral tonic spasms. Less commonly, blepharospasm manifests as recurrent clonic spasms and even less frequently can present as eyelid apraxia [12]. The onset of symptoms is often subtle with a variable rate of progression on the order of weeks to months alternating with periods of stability. It is usually bilateral at onset, but it can begin unilaterally. Unilateral onset represents approximately 25% or less of cases [1]. Spontaneous remission occurs in approximately 10% of patients and is often incomplete with subsequent exacerbation [12]. 

Focal dystonias, like other dystonias, are often activated by movement of the affected muscles and exacerbated by anxiety. Some cases exhibit a sensory trick, or geste antagoniste, which is a phenomenon where sensory stimuli applied to the affected or nearby body region briefly lessens the dystonic contraction. The most common sensory tricks for cranial dystonia, especially blepharospasm, are touching the face or eyelid, humming, singing, or talking. Some studies report that up to 87% of people affected by blepharospasm can have one of these sensory tricks [13], but our experience finds this percentage to be much lower in clinical practice. 

#### 3.3.2. Hemi-Facial Spasm

In contrast to blepharospasm, hemi-facial spasm (HFS) is not a dystonia, but rather a peripheral myoclonus, and is characterized by unilateral orbicularis oculi contraction that can also involve lower facial muscles. HFS is considered primary when due to an ectatic blood vessel near the exit zone in the brainstem and secondary when due to damage to the brainstem or facial nerve from any other cause such as stroke, infection, trauma, tumor, or inflammatory condition such as Bell’s Palsy. Most cases are primary in a 4:1 ratio. HFS may very rarely occur bilaterally and is distinguished from blepharospasm by the presence of asynchronous spasms [14].

#### 3.3.3. Oromandibular Dystonia

Oromandibular dystonia (OMD) involves involuntary tonic or clonic muscle contractions in differing combinations of lower facial muscle groups including masticatory, pharyngeal, labial, or tongue (lingual dystonia) [15]. OMD causes difficulty chewing, talking (tongue or dysphonia), and/or swallowing. Masticatory muscle involvement is the most common form of OMD and causes three main phenomenologies: jaw clenching, jaw opening, or jaw deviation in descending order of occurrence. It can occur in isolation, but is often associated with another focal dystonia affecting a contiguous body region such as eyes, platysma, neck (cervical dystonia), or larynx (spasmodic dysphonia) [16]. Talking and chewing often exacerbate OMD, and when severe, can cause difficulty speaking, eating, and drinking. OMD is often associated with temporal mandibular joint pain and oral injury. 

#### 3.3.4. Meige Syndrome

Meige syndrome, or cranial-facial dystonia, is the combination of blepharospasm and oromandibular dystonia. The first potential recognition of the syndrome was through Pieter Brueghel the Elder’s 16th century painting of “De Gaper” or “Yawning Man”. Dr. C.D. Marsden credits R.E. Kelly for first recognizing this depiction and suggests naming the condition Brueghel’s syndrome [15]. Meige and Feindel, however, were the first to clinically describe this syndrome in 1910 in their work Les Tics Et Leur Traitment [2]. 

Meige syndrome can be a primary condition or associated with a neurodegenerative process. Some pathological data have shown basal ganglia Lewy body deposits. It can also be associated with other movement disorders, such as essential tremor or parkinsonism [17,18].

## 4. Blepharospasm and Oromandibular Dystonia: Treatment 

Blepharospasm and oromandibular dystonia have several treatment options including BoNT therapy, medication, and surgical intervention. BoNT is widely accepted as a first-line therapy.

### 4.1. Non-Toxin Therapies

Medications, such as anticholinergics (trihexyphenidyl and benztropine), benzodiazepines, VMAT2 inhibitors (tetrabenazine), levodopa, and baclofen, have been used with variable success. While oral medications have limited efficacy, anticholinergics and benzodiazepines are the most likely to have some benefit [19]. Medication efficacy is at best modest and does not show the same level of efficacy when compared to BoNT. Oral medication therapies are further limited by systemic side effects which are not usually seen with botulinum toxin therapy. Additionally, benzodiazepine use is complicated by possible tolerance and addiction. 

Deep brain stimulation (DBS) of the internal portion of the globus pallidum is efficacious for both segmental and generalized dystonias. DYT1 positive dystonias appear particularly sensitive, but efficacy has been shown for DYT1 negative forms as well [20]. One example is BoNT-resistant cervical dystonia, especially when dynamic in nature. DBS has not been extensively studied in blepharospasm, OMD, or Meige syndrome and is therefore, not a first-line therapy. There is emerging evidence that DBS may be helpful in select patients with BoNT-resistant cranial dystonias [21]. 

### 4.2. Botulinum Toxins

In 1820 Dr. Justinus Kerner, a German physician, described the symptoms of botulism caused by consuming smoked sausage and postulated the potential medical use of the toxin to aid in the treatment of muscle hyperactivity caused by “hyperexcitability of the sympathetic nervous system”. The responsible bacteria was first isolated by Dr. Emile van Ermengem, professor of bacteriology at the University of Ghent, in 1897 from a ham that had claimed the lives of 3 of 34 affected Belgian musicians. He named this bacteria *Bacillus botulinum*, which would be later called *Clostridium botulinum* [22]. 

Dr. Alan Scott, an ophthalmologist, first described the effect of BoNT on extraocular movements in monkeys affected by strabismus in 1981 [23]. This ultimately led to the FDA approval of the first BoNT for the treatment of strabismus, blepharospasm, and hemi-facial spasm in 1989. Since that time, there have been two additional preparations of type A and one type B BoNT approved by the U.S. Federal Drug Administration (FDA). Clinical uses of BoNT have greatly expanded over the past three decades and this has created a long track record of safety and efficacy. While, there are a number of trials that look at individual toxins for each indication, there are a limited number of prospective randomized control trials and only a few prospective controlled head-to-head studies. This prevents the highest level of validation for some observed efficacy trends.

BoNT is a common first-line therapy for the treatment of many dystonic syndromes. A number of large safety studies have shown efficacy and no significant long-term side effects. Levels of evidence for blepharospasm and OMD are summarized in Table 1 [24]. 

The 2016 AAN guidelines updated levels of evidence for all toxins. This dropped the level of evidence for all toxins for these dystonias with the exception of rimaBoNT/B that was already level U for both dystonias [24]. One explanation for this drop in evidence was posed by Jankovic and colleagues in 2017 by referencing the dearth of large randomized trials looking at toxins for each indication [26]. Despite these new guidelines the toxins with the highest levels of evidence remained the same and the standard of care has not been changed. OnaBoNT/A and incoBoNT/A carry the highest recommendations for blepharospasm. OnaBoNT/A and aboBoNT/A carry the highest evidence for oromandibular dystonia. Table 1 summarizes the levels of evidence for each toxin.

### 4.3. Injection Technique and Site Selection

The general approach for first-time BoNT injections, especially in those patients that are naïve to BoNT, should focus on using minimum effective starting doses. This helps to prevent side effects such as excessive focal muscle weakness. Table 2 details common starting doses for each toxin with the highest level of evidence for each focal dystonia. There is consensus in the starting doses for each toxin, but the maximum doses used within each dystonic syndrome is variable, and therefore, omitted. 

While there are standard muscle sites used, each injection session should be tailored to the individual patient. Threshold levels for effect and weakness are different for each person and there is no direct correlation between dose and level of effect or duration. Anecdotally, there is a therapeutic window within which there is effect without side effects and increasing the dose slightly within this window has the potential to improve duration of effect. However, this change in duration is often not significant in our experience.

The preparation of incoBoNT/A or onaBoNT/A must include a reconstitution with preservative free 0.9% sterile saline. While several differing dilutions could be utilized, a typical dilution is 1 mL per 100 U (units) vial resulting in a concentration of 10 U/0.1 mL. Details on additional dilutions are provided in the package inserts for each toxin and would be necessary for attaining precision in administering small single unit and fractional quantities. BoNT diffuses through fascia and adjacent muscles. Differences in concentration of BoNT may affect diffusion into adjacent muscles but does not appear to make a difference with regards to efficacy [27]. Clinicians should weigh the pros of dilution, such as dose resolution with the cons such as introducing larger fluid volumes into a small compartment.

Per onaBoNT/A and incoBoNT/A product labeling, injections may be performed using a 27-gauge to 32-gauge needle without electromyography (EMG) guidance. However, common practice is to use the smaller 0.5-inch 30-gauge to 32-gauge needles to minimize pain and bruising. For BoNT naïve patients, initial site dose is recommended to start at 1.25 U to 2.5 U per site. However, in common practice minimal side effects are seen when starting at 5 U per site. Many practitioners begin with the 4 to 5 sites in the orbicularis oculi and surrounding muscles involved listed in Figure 2. Orbicularis oculi is targeted first with injections directed to the most active muscles. Palpation of the orbital ridge is a helpful landmark for the injection sites because it makes them more independent of facial shape and subcutaneous tissue. Care should be taken to avoid the *midline* of the upper lid and brow in order to avoid ptosis caused by weakening the levator palpebrae superioris. However, injecting palpebral sections of the orbicularis oculi near the levator is sometimes necessary in cases where the benefit is insufficient when using this initial approach. Levator is shown as a shadowed muscle underneath orbicularis oculi in Figure 2. Similarly, avoiding the *medial* lower eyelid near the nasal bridge aids in reducing the possibility of chronic tearing caused by weakness of the musculature that keeps the inferior punctum (tear duct) approximated to the globe and diplopia caused by diffusion into the inferior oblique. Doses greater than 10 U per site do not typically produce superior efficacy and total dose per treatment session exceeding 200 U total (100 U per side) is not common [28].

OMD injections are prepared similarly and also routinely use a 30-gauge needle for superficial muscles and a 27-gauge EMG needle for deeper muscles that are difficult to visualize, such as the lateral pterygoids and genioglossus. Entry into these facial muscles is directed with the needle perpendicular to the skin. Figure 3 shows the most commonly injected muscles for OMD. In most cases, the masseter is the muscle first considered, and then muscles are added as needed based on their perceived level of activity [29]. Care should be taken to avoid injection of the parotid gland that overlies the posterior border of the masseter to prevent xerostomia. Injection of the lateral pterygoid using EMG guidance is directed either intra-orally or externally through the mandibular notch (incisura) which is located 2 to 4 cm anterior from the external auditory canal at about the level of the tragus and about 1 cm below the inferior margin of the zygomatic arch. Insert the needle with a slight upward trajectory of approximately 15 degrees through the notch while instructing the patient to translocate the jaw contralaterally will provide EMG feedback [30]. 

Injection of the genioglossus to reduce tongue protrusion can be done from a submandibular approach with EMG guidance (Figure 2). Genioglossus injections are generally 2.5 cm to 3 cm posterior from the chin about 1 cm on either side of the midline approximately 2 cm apart from each other. These injections are medial to the digastric injections. They should be at least 1 cm into the tissue because the genioglossus is deep to the subcutaneous tissue as well as the mylohyoid and geniohyoid [31]. Injection into the orbicularis oris to reduce lip movement or pursing is problematic due to cosmesis concerns for potential mouth asymmetry, difficulty drinking thin liquids, and drooling caused by focal weakness. The starting doses for each muscle group are summarized in Table 2 and scaled with respect to onaBoNT/A. The starting doses for other toxins can be determined by using consensus dosage conversions. The approximate conversion from onaBoNT/A to incoBoNT/A, aboBoNT/A, or rimaBoNT/B is calculated by multiplying units of onaBoNT/A by 1, 2.5, and 50, respectively [16,32].

### 4.4. EMG-Guided Injections

EMG and ultrasound can be used to help guide BoNT injections and increase accuracy of muscle selection which maximizes benefit while minimizing potential side effects. The benefit provided by EMG is not substantial for some easily targeted muscles, but is essential in targeting muscle groups that are difficulty to directly visualize or palpate such as lateral pterygoid and genioglossus for OMD. EMG is generally not required for injection of facial muscles, especially for blepharospasm. 

### 4.5. Assessing Benefit

Treatment response is usually assessed through patient report and clinical assessment. Several rating scales exist for dystonias such as blepharospasm and OMD. Many of the scales are either subjective, questionnaire-based or time-consuming clinician-applied examination [36]. Ratings scales mainly exist to provide more standardized assessments in clinical trials and are not routinely used in clinical practice. 

### 4.6. Potential Side Effects and Precautions

Side effects of BoNT therapy are centered on the main mechanism of action. The most common side effect is excessive focal muscle weakness. When muscle selection is chosen carefully, and dosing started conservatively, excessive weakness is minimized. The two most common forms of weakness with blepharospasm are ptosis and lagophthalmos. Eye dryness and diplopia are less common adverse events. The most common adverse events experienced when treating OMD are chewing weakness, dysphagia, dysarthria (tongue injections), and dry mouth (diffusion into salivary glands). All BoNT labeling includes a boxed warning concerning spread of toxin effect. They also include rare side effects, such as generalized weakness, allergic reactions, and flu-like symptoms [28].

While BoNT is a first-line therapy, clinicians should consider using oral medications as primary therapy when relative contraindications exist, such as pre-existing neuromuscular conditions causing weakness in areas under consideration for injection. Local skin infections or recent febrile illness should also postpone injections and could warrant a trial of oral medication. 

### 4.7. Future Developments

A new BoNT preparation, daxibotulinumtoxinA (RT002; Revance Therapeutics, Inc., Newark, CA, USA), is in phase 3 clinical trials for cervical dystonia, spasticity, and plantar fasciitis. Preliminary data from phase 2 study in cervical dystonia showed similar efficacy to other toxins with a potentially longer duration of effect [37].

## 5. Conclusions and Recommendations

Blepharospasm and OMD are common focal dystonias and are often seen in isolation, but can be seen together as a segmental dystonia called Meige syndrome. Practice guidelines list BoNT therapy as a first-line treatment. OnaBoNT/A has a 30-year treatment history for blepharospasm and OMD with well-established safety and efficacy. Additional BoNTs are now available and more are under development. There are limited numbers of prospective randomized, placebo-controlled studies looking at each toxin for each focal dystonia. Performing these studies now would be challenging and costly. 

Dystonia is a dynamic entity. Re-assessing the most active muscles improves targeting, ensures the greatest efficacy, and minimizes side effects. There are four key aspects for successfully using BoNT injections for blepharospasm and OMD that should be used at each injection session: Educate patients about expected outcomes and possible side effects.Reassess the predominant phenomenology and most active muscles.Modify the injection regimen to include only the most active muscles.Reduce or eliminate injections that have previously caused side effects.

## Figures and Tables

**Figure 1 toxins-12-00269-f001:**
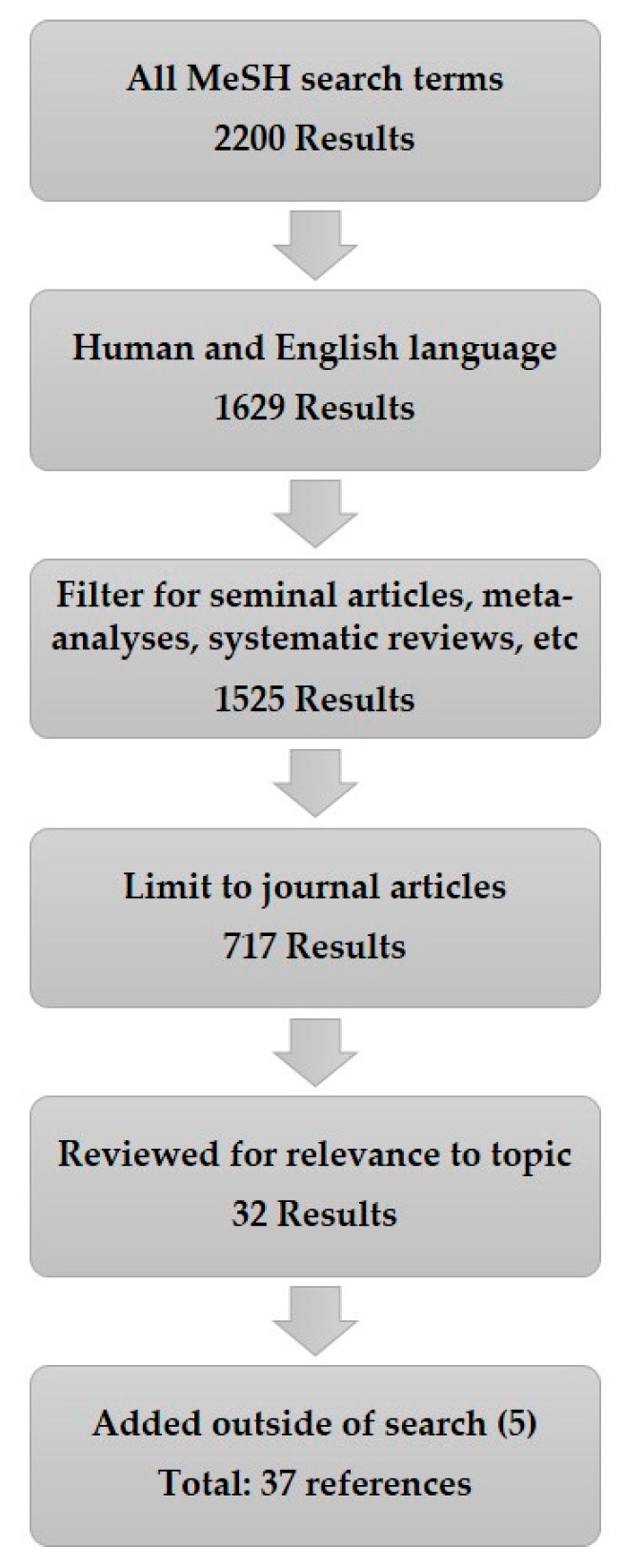
Literature review process.

**Figure 2 toxins-12-00269-f002:**
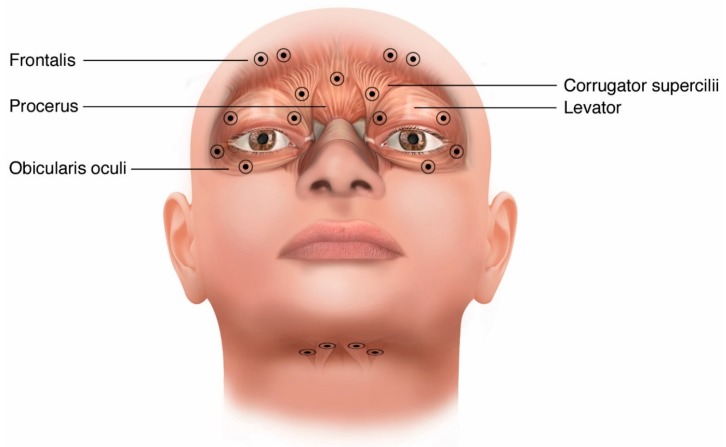
Injection site options for blepharospasm and tongue injections for OMD.

**Figure 3 toxins-12-00269-f003:**
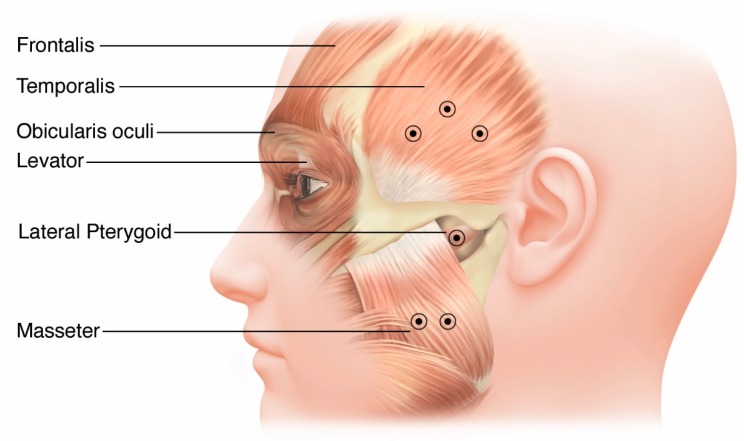
Anatomy and site selection options for oromandibular dystonia.

**Table 1 toxins-12-00269-t001:** Levels of Evidence for Botulinum Treatment of Focal Dystonia.

Botulinum Toxin (Strain)	Focal Dystonia Subtype	Level of Evidence
Onabotulinum (A)	Blepharospasm	B
	Oromandibular dystonia	C
Abobotulinum (A)	Blepharospasm	C
	Oromandibular dystonia	C
Incobotulinum (A)	Blepharospasm	B
	Oromandibular dystonia	U
Rimabotulinum (B)	Blepharospasm	U
	Oromandibular dystonia	U

A—Effective; B—Probably Effective; C—Possibly Effective; U—Insufficient Evidence Adapted from [25].

**Table 2 toxins-12-00269-t002:** Common Muscles and Dosing for Blepharospasm and Oromandibular Dystonia.

Focal Dystonia	Common Muscles Injected	Common Starting Doses (U) (onaBoNT/A, incoBoNT/A) *
Eye Closure	Orbicularis Oculi	20 to 25
Expression	Corrugator	10
	Procerus	5
	Frontalis	20
Jaw Closure	Masseter	50
	Temporalis	40
	Medial Pterygoid	20
Jaw Opening	Lateral Pterygoid	20
	Anterior Belly of Digastric	5 (per belly)
Jaw Deviation	Contralateral lateral pterygoid	20
	Ipsilateral temporalis	40
Tongue Protrusion	Genioglossus	20 (10 each side)

* IncoBoNT/A has been studied and shown efficacious for blepharospasm but data is limited for use in OMD. Dosing data derived from multiple sources [28,33,34,35].

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
