# Peer review of "Treatment of Blepharospasm and Oromandibular Dystonia with Botulinum Toxins"

_toxins, 2020, doi:10.3390/toxins12040269_

Round 1

Reviewer 1 Report

This is a very well written and comprehensive Review.

From my point of view, there should be added one short paragraph about the diagnostic way (technical

investigations, ...).

Author Response

Thank you for your review.  It seems part of your suggestions were cut off.

Can you clarify the following:

...one short paragraph about the diagnostic way (technical investigations, ...).

Diagnostic way of what?  I don't follow.

Thank you.

Reviewer 2 Report

This is an interesting, generally well-written review; however, it should be re-checked carefully for typos (e.g. p.5, line 208: “… Care should be taken TO avoid …”; and several other similar typos) and in a few instances style and grammar.

Although the sections on e.g. classification, epidemiology and history of botulinum toxin treatment are well readable, in my opinion they do not add relevant new information to the existing literature, but rather provide another review of existing studies and historical developments that have been well documented and reviewed previously.  

Under “injection technique and site selection”, a comparatively frequent problem encountered in clinical practice in treating patients with blepharospasm with botulinum toxin is not addressed in the current version of this manuscript: in addition to the injection sites for the M. orbicularis oculi indicated in Fig. 1, for some patients pretarsal injection sites (palpebral part of the M. orbicularis oculi) may be required if the therapeutic effect is otherwise insufficient (see e.g. Reichel et al., Akt Neurol 2009), although injecting BoNT in these pretarsal/palpebral injection sites does carry a higher risk of side effects (mainly ptosis). This issue should be addressed respectively discussed by the authors.  

Specific (mostly minor) comments:

  1. “non-toxin therapies”, p. 4, line 146: “… is efficacious for DYT1 dystonia and BoNT resistant cervical dystonia. …”: DBS has been shown to improve not only DYT1-positive segmental or generalized dystonia, but also other forms of segmental and generalized dystonia (“DYT1-negative”). This should be considered by the authors and a corresponding statement be added.
  2. “injection technique and site selection”, p. 5, line 197: “…Differences in concentration of BoNT may affect diffusion into adjacent muscles but does 197 not appear to make a difference with regards to efficacy.” If possible respectively available, evidence for this statement should be cited.
  3. Table 2: in my opinion, common starting doses for abobotulinumtoxinA should be included in this table, possibly also for rimabotulinumtoxinA.

Author Response

Thank you for your thoughtful review.

To specifically address your comments I did the following: 

  1. Addressed obvious typos and grammatical errors where found.
  2. Corrected reference to DBS and dystonia to include DYT1 negative segmental and generalized dystonias with reference. Lines 149-150
  3. Injection technique and site selection: Comment regarding differences in concentration affecting spread and affect has been modified with new reference
  4. Starting doses for all toxin types were addressed in the text near Table 2 rather than adding all to Table 2. This was done for retaining visual appeal and simplicity.
  5. Addressed concern for palpebral segment of orbicularis with modification of text. New lines 217-219

Reviewer 3 Report

In this manuscript, the authors present a comprehensive review of botulinum toxin use in blepharospasm and OMD. The section of injection guidance is very well written. The pictoral presentation and tables are excellent and this manuscript would made a good reading for fellows starting in movement disorders. I have the following suggestions that I believe could further improve the paper.

  1. The methodology of the paper is well described (line 44-49). I would suggest the authors present the data in a visual format (as a flowchart). This would add to the paper.

  1. Consider adding the following aspects to the review:

a) Literature on the psychiatric aspects of focal dystonias. Reference: Berman BD, Junker J, Shelton E, et al. Psychiatric associations of adult-onset focal dystonia phenotypes. J Neurol Neurosurg Psychiatry. 2017;88(7):595–602. doi:10.1136/jnnp-2016-31546

b) The presence of a reported “central benefit” of botulinum toxin in dystonia by way of sensory afferents/ feedback and sensorimotor integration. This has been shown using MEG in cervical dystonia (Reference: Mahajan A, Alshammaa A, Zillgitt A, et al. The Effect of Botulinum Toxin on Network Connectivity in Cervical Dystonia: Lessons from Magnetoencephalography. Tremor Other Hyperkinet Mov (N Y). 2017;7:502. Published 2017 Nov 10. doi:10.7916/D84M9H4W, Mahajan A, Zillgitt A, Alshammaa A, et al. Cervical Dystonia and Executive Function: A Pilot Magnetoencephalography Study. Brain Sci. 2018;8(9):159. Published 2018 Aug 22. doi:10.3390/brainsci8090159) and more recently, in blepharospasm. ( Reference: Jochim A, Li Y, Gora-Stahlberg G, et al. Altered functional connectivity in blepharospasm/orofacial dystonia. Brain Behav. 2017;8(1):e00894. Published 2017 Dec 18. doi:10.1002/brb3.894)

  1. Section 4.3. I believe the authors are referring to Table 2 instead of Table 1.

Author Response

Thank you for your thoughtful review.

To specifically address your comments I did the following: 

  1. Addressed methodology with new flowchart figure. Now Figure 1.
  2. My mandate from my editor for this portion is not to mention mechanisms of action as this is a special edition and there is another section dedicated to that.  I am afraid adding these might cause this manuscript to stray to far from that mandate.
  3. I have fixed the reference regarding Tables.
  4. There are few other changes made based on comments from other reviewers.  The new "edited" manuscript is attached.

Regards,